# Management of Post-Harvest Anthracnose: Current Approaches and Future Perspectives

**DOI:** 10.3390/plants11141856

**Published:** 2022-07-15

**Authors:** Alice Ciofini, Francesca Negrini, Riccardo Baroncelli, Elena Baraldi

**Affiliations:** Department of Agricultural and Food Sciences (DISTAL), University of Bologna, Viale Fanin 44, 40126 Bologna, Italy; francesca.negrini6@unibo.it (F.N.); riccardo.baroncelli@unibo.it (R.B.)

**Keywords:** crop protection, anthracnose, *Colletotrichum*, post-harvest

## Abstract

Anthracnose is a severe disease caused by *Colletotrichum* spp. on several crop species. Fungal infections can occur both in the field and at the post-harvest stage causing severe lesions on fruits and economic losses. Physical treatments and synthetic fungicides have traditionally been the preferred means to control anthracnose adverse effects; however, the urgent need to decrease the use of toxic chemicals led to the investigation of innovative and sustainable protection techniques. Evidence for the efficacy of biological agents and vegetal derivates has been reported; however, their introduction into actual crop protection strategies requires the solutions of several critical issues. Biotechnology-based approaches have also been explored, revealing the opportunity to develop innovative and safe methods for anthracnose management through genome editing and RNA interference technologies. Nevertheless, besides the number of advantages related to their use, e.g., the putative absence of adverse effects due to their high specificity, a number of aspects remain to be clarified to enable their introduction into Integrated Pest Management (IPM) protocols against *Colletotrichum* spp. disease.

## 1. Epidemiology and Pathology of Colletotrichum spp.

### 1.1. Interaction between Colletotrichum spp. and Their Hosts

The genus *Colletotrichum* comprises more than 200 fungal species, informally gathered in 15 species complexes [1]. Many of them are pathogens of important crops where they cause anthracnose, a severe disease with great economic impact. For this reason, *Colletotrichum* spp. have been listed among the top ten most relevant fungal pathogens worldwide [2,3].

Plant infections occur mainly in tropical and sub-tropical regions and less frequently at temperate latitudes since their onset requires warm temperature and high relative humidity [4,5]. Duration and intensity of rainfalls, wetness of the leaf surface and light intensity have also been reported as factors positively correlated with the infective process [6]. For these reasons, crops located in regions with frequent precipitations, never really drying between rainfalls, are particularly affected by *Colletotrichum* infections.

With respect to fruit infection, the early stages of the interaction with host tissues are similar for all the *Colletotrichum* species [7]: conidiospores spread from infected vegetal material or through insects, adhere by means of a hemicellulosic mucilage to the external vegetal surface, germinate and infect often by mean of specialized structures, such as appressoria [8,9,10,11]. Infections can take place even by penetration through stomata, lenticels, wounds, or abscission of scar tissue [11,12,13,14,15]. Although specific host–pathogen interaction and infection strategies have been detected, such as in *C. acutatum sensu lato* that exhibit four different colonization pathways [16], the process proceeds according to two main strategies, depending on the different species, hosts, and tissues: (i) intracellular hemibiotrophy or (ii) subcuticular, intramural necrotrophy [6,7,17,18]. The first one includes an initial short (from 24 h to over 72 h) biotrophic stage, characterized by the formation of intracellular primary hyphae, largely differing among species in their morphology. This stage is followed by a destructive necrotrophic phase when secondary ramified narrow hyphae spread throughout the host tissues [19,20,21,22]. “Signals” from the fruit ripening stage, such as occurrence of disaggregated cell wall components, accumulation of ethylene, changes in the environmental pH and in the content of organic compounds, decrease of both antifungal substances (e.g., polyphenols and phytoalexins) and host defense mechanisms can act as promoters of the pathogen necrotrophic phase [23,24,25,26,27]. On the contrary, subcuticular, intramural necrotrophy, typical of *C. capsici* (syn. *C. truncatum*) and *C. circinans*, consists of an early asymptomatic short (24 h) stage where the pathogen grows within the periclinal and anticlinal walls of the epidermal cells (intramural development). Subsequently, these species rapidly spread throughout the host tissues producing destructive effects [28,29].

### 1.2. Antracnhose Disease: A Challenge for the Agri-Food Sector

Anthracnose symptoms highly vary depending on the plant tissue and cause huge losses of fruit production [11]. For instance, strawberry plants, which are susceptible to several *Colletotrichum* species, can exhibit serious anthracnose symptoms in all its parts [30,31]. For this crop, anthracnose can account for up to 80% and more than 50% of plant and fruit losses in nurseries and in field, respectively [31,32,33].

In general, symptoms on stems or leaves range from small greyish-brown sunken spots to darker patches on branches. On the contrary, infected blossoms appear dry and rotten with compromised fruit development, whereas on fruits anthracnose produce dark lesions surrounding pink or orange conidia masses that under suitable conditions can evolve causing tissue necrosis [4,11,34,35,36]. In any case, the highest impact of this disease is due to the fruit damage during postharvest storage (Figure 1). 

Here symptoms can cause consistent decays and a reduction in both quality and aesthetic standards, resulting in severe economic losses [23,37]. For instance, banana fruits are highly affected by anthracnose that is responsible for losses up to 30–40% of the marketable products [38]. The damages caused by this disease even grow in other susceptible hosts. In this context, the shelf life of papaya in India, the main producer of this fruit worldwide [5], is seriously compromised by *Colletotrichum* spp., determining losses up to 93% [39,40]. Similarly, anthracnose represents a severe post-harvest disease for mango fruits, especially in regions where infections are favored by the climatic conditions and its incidence can reach the 100% of the production [23]. 

Post-harvest anthracnose is mainly due to infections taking place in the field that become quiescent until the occurrence of conditions favorable for pathogen germination and development [37,41,42]. As a result, the disease is particularly severe for climacteric fruits such as banana, guava, avocado, pear, mango, and papaya since their ripening stage includes physiological and biochemical changes, due to the re-modelling and degradation of cell walls, providing suitable conditions for the fungal development [11,43,44]. Nevertheless, anthracnose represents a serious constrain even for non-climateric fruits, such as strawberry, citrus, and dragon fruit [3,45,46,47].

After a brief insight into the methods traditionally applied for the management of anthracnose disease, this review will revise the most important alternative and eco-friendly approaches that have been recently explored to counteract anthracnose, with a special focus at their possible application at post-harvest stage. Particular attention will be given to recently developed biotechnology-based strategies, that offer high specificity and low risk of negative effects on environment and human health (Figure 2).

## 2. Traditional Approaches

The strategies traditionally applied to counteract anthracnose disease in fruits after harvesting are mainly based on physical treatments and applications of few chemical fungicides.

Physical treatments include environmentally friendly practices finalized to inhibit *Colletotrichum* development, such as vapor heat, forced-air dry heat and hot water dips of harvested products. However, some of these storage techniques are not suitable for the majority of fruit production, since potentially affecting the fruit quality. For instance, hot water dips have been associated to decays of organoleptic and nutritional features and subsequent reduction of shelf life [3,48,49,50]. 

On the other hand, chemical-based control of anthracnose (both in the field and in commercial packinghouses after harvest) has been for many years the main tool to prevent the damages. Indeed, synthetic fungicides are effective means to reduce *Colletotrichum* inoculum in the field and the fungal development during the fruit storage [3,15]. The products traditionally used against *Colletotrichum* spp. include cupric products, strobirulines, dithiocarbamates, benzimidazole, and triazole compounds, together with other chemicals, such as prochloraz, imazalil, and chorothalonil [3,7]. However, their improper use, as well as consecutive treatments based on substances with the same mechanisms of action, have been related to decreases in the sensitivity of pathogens and the emergence of resistant strains [23,33,47,51,52,53,54,55,56,57,58,59,60]. For instance, impaired sensitivity toward prochloraz and benzimidazole has been revealed in *C. gloeosporioides sensu lato* isolates from avocado and mango fruits [23,51]. In addition, the use of fungicides in the agri-food sector represents a serious risk for the human health and the environment, given both the presence of chemical residues on fruits and their pollutant effects on soil, water, and not-target organisms [3,55,61,62,63,64,65]. For example, the use of thiophanate-methyl in post-harvest anthracnose control, has been associated with severe toxic effects on the human health [3]. Similarly, prochloraz, effective to prevent the phytopathology on mango and avocado fruits [66,67], has been recently listed as a priority pollutant by the US Environmental Protection Agency (EPA) because of its putative carcinogenic effects. 

Recently several Governments introduced regulations on the use of phytosanitary products in the agri-food sector and the presence of chemical residues on final products (Maximum Level Residues, MLR), such as the European directive 2009/128/CE (http://data.europa.eu/eli/dir/2009/128/o, accessed on 21 December 2021) that was deliberated with the aim to mitigate the adverse effects of the agricultural practices and promote the implementation of safer approaches in the European Union. As a consequence, some fungicides used to counteract *Colletotrichum* spp. have been banned [68].

Restrictions vary among countries in both the categories of active compounds regulated and the MLRs allowed [69]. In this context, farmers of exporter countries importing in countries with higher standards incur in great investments for the adjustment of their productive systems, such as expensive inputs and specialized human capital. According to Fiankor et al. [69], many producers not able to support the expenses for this renovation process are bounded to exit the market.

On the basis of these considerations, it emerges the urgency of a transition toward innovative, effective, and sustainable strategies; indeed, over the years many efforts have been made worldwide to find alternative tools to reduce the impact of the traditional fungicides. In some cases, the research has been oriented toward antifungal active substances with low toxicity profiles, such as fludioxonil [66,67], but scientific investigations have mainly focused on the development of chemical-free managing approaches and technologies.

## 3. Innovative and Sustainable Approaches

### 3.1. Biological Strategies

#### 3.1.1. Biological Control

Biocontrol-based approaches can represent a valid strategy to improve sustainability in agriculture. These techniques are based on the antagonistic activity exerted by bacterial, fungal or yeast species that can be used to develop non-polluting commercial formulates. The mechanisms by which they act are various and include space and/or nutrient competition, parasitism, and production of toxic metabolites [70]. 

Post-harvest management of anthracnose based on biological control strategies are available both in the field, to prevent infections, and after the harvesting, to limit the pathogen development [70,71,72,73]. 

In recent years the efficacy of this approach has been largely explored, finding several biological agents (or antifungal compounds extracted from them) potentially suitable to counteract *Colletotrichum* spp. infection or development in susceptible fruits (Table 1). 

Although many species are responsible for consistent damages, investigations were focused mainly on *C. acutatum* and *C. gloeosporioides* species complexes and *C. truncatum* [73,85,86,87]. In this regard, some studies revealed the strong efficacy of bacterial species, such as some strains of *Bacillus subtilis* and *Paenibacillus polymyxa*, and their derivates to reduce both the incidence of anthracnose caused by *C. acutatum* and/or *C. gloeosporioides* species complex and the severity of the post-harvest lesions [73,74,75,76]. In particular, *Bacillus* spp. were shown as prominent candidates for the biological control of these pathogen species in various host species since capable to highly reduce anthracnose incidence (from 76% to 83%) and severity (from 65% to 85%) [74,75].

Moreover, *Streptomyces philanti* has been suggested as a promising tool for the biological control of *C. gloeosporioides* species complex, since treatments with volatile compounds produced by this species resulted in a complete absence of symptoms in chili fruits [77]. Some strains of *Burkholderia* spp. and *Pseudomonas aeruginosa* exhibited instead a strong efficacy for the control of *C. truncatum* in chili, as reported by Sandani et al. [78] that revealed a reduction of anthracnose incidence ranging from 75% to 100%.

Moreover, various filamentous fungi have been suggested as efficient biological tools for post-harvest anthracnose prevention; in this case, the antagonistic activity can be also due to the capability of these microorganism to colonize *Colletotrichum* hyphae [73]. With respect to this, *Trichoderma* spp. have been shown very effective [79,80,81]; indeed, investigations conducted on strawberry [79] and banana [81] revealed a great efficacy of these fungi to reduce anthracnose incidence (−51%) and severity (−88%) respectively. Furthermore, Oliveri et al. pointed out also that the transformation of citrus plants with a *T. harzianum* gene, encoding an antifungal protein, resulted in decreased anthracnose symptoms in fruits inoculated with *C. gloeosporioides* species complex [80]. 

Finally, also various yeast species were shown to be excellently efficient *Colletotrichum* antagonists [73], as in the case of *Metchnikowia pulcherrima* and *Pichia kluyveri*, more effective than any other tested bacterial or fungal species in contrasting *C. acutatum* species complex infections in apple fruits [82,83,84]. Here, the biological control exerted by these species allowed a 100% reduction of anthracnose incidence. The efficacy of yeast species against *Colletotrichum* infections was even reported in some field trials. For instance, the application of a *Rhodotorula minuta* suspension of a mango orchard was revealed even more effective of chemicals in controlling anthracnose disease [86].

Despite this scientific evidence, to date, only two microorganism-based biofungicides against *Colletotrichum* spp. are commercially available, both containing bacterial strains belonging to the genus *Bacillus* [73]. The first one is Serenade ASO (composed by *B. amyloliquefaciens* QST713, former classified as *B. subtilis* QST713), whereas the other is Double Nickel 55 (containing *B. amyloliquefaciens* D747), commercialized by Bayer and Certis respectively. Notably, whereas Serenade ASO has been registered even in the European Union (EU), Double Nickel 55 is currently not commercially available in the EU (while its commercialization has been authorized in other countries such as Canada and USA). However, as reported above and underlined by Shi et al. [73], other microorganisms perform greatly in anthracnose control, suggesting that more effort is needed to provide farmers with more biological tools.

#### 3.1.2. Plant Derivates

The research of eco-friendly methods to manage anthracnose disease has been often oriented toward plant derivates with direct antifungal activity (e.g., secondary metabolites involved in the plant immunity mechanisms). 

Over the years, the potential of several plant-based substances against *Colletotrichum* spp. has been examined, by research institutions in countries where anthracnose represents a severe economic threat. However, many studies have limited their investigations to the inhibitory activity exerted by some crude vegetal extracts on cultured fungi without exploring their efficacy against anthracnose on fruits i.e., [88,89,90,91,92]. Therefore, their possible application in post-harvest managing strategies is currently scarcely known. Moreover, these substances need to be carefully evaluated not only to define their fungicide potential but also to exclude any undesirable adverse effects of their use. With respect to this, Bordoh et al. [92] pointed out a dose-dependent phytotoxicity and a deterioration of some organoleptic features in dragon fruits following treatments with two crude ginger extracts that were found effective against *C. gloeosporioides* species complex. 

Research conducted on papaya fruits revealed that extracts purified from *Vitex mollis* pulp can exert enhanced antifungal activity compared to the crude ones suggesting that partitioned fractions of plant derivates can be preferable for pathogen management [93]. A class of purified extracts that is receiving increasing attention as fungicide alternatives is represented by essential oils (EOs) from several aromatic plants [94]. Indeed, EOs exhibit a strong antifungal activity related mainly to two categories of lipid active compounds: terpenoids and phenylpropanoids, capable to interact with hydrophobic components of the pathogen membrane [95,96,97,98]. EOs are known from ancient times for their antimicrobial activity, and in recent years, they have been largely explored to assess their suitability to replace chemicals in agriculture. 

The effectiveness of some of these substances for the post-harvest management of anthracnose has been deeply studied and shown effective for different fruits [97] (Table 2). 

Investigations carried out on papaya fruits showed that the development of *C. gloeosporioides* species complex is affected by different EOs such as lemongrass, ginger, savory and thyme oils [99,100,101]; in particular, the latter was shown the most effective to reduce the anthracnose incidence (−26.5%) [101]. 

Savory and thyme oils have been demonstrated effective in limiting *C. gloeosporioides* species complex growth also in avocado fruits [102] where both the incidence and the severity of the disease decreased by the application of the EOs; whereas in mango, only thyme oils showed to act against these species [103]. EOs extracted from thyme, as well as cinnamon bark oils, were also efficient in reducing the incidence of anthracnose disease caused by *C. acutatum* species complex on strawberry fruits [104]. On the contrary, *C. musae* was shown to be sensitive to the application of EOs extracted from *Ocimum basilicum* and *Ocimum gratissimum* with reduced anthracnose severity on banana fruits [105]. 

Nevertheless, also for this class of compounds, knowledge on undesirable effects of EOs is scarce although fundamental. For example, Ali et al. [99] found a dose-dependent phytotoxicity on papaya fruits treated with lemongrass oils. 

On the other hand, the suitability of various formulations to confer stability and durability to EOs, which display high volatility and hydrophobicity has received great attention. Nanotechnology-based methods, such as EO encapsulation, incorporation into edible or biodegradable coatings, and development of microemulsions, have been suggested as promising candidates to achieve stabilized formulates [96,106,107,108,109,110,111,112].

Among the plant active compounds explored as alternative solutions for crop protection, several algal derivates have been found to act as plant resistance inducers against biotic and abiotic stressors. Seaweed-based commercial formulates have been developed for young seedling immersions or foliar (high/low-pressure) spraying treatments. In addition, in vitro studies using cultured fungi revealed that these substances can also display inhibiting effects on the mycelial growth and/or the conidial germination of pathogenic agents [113]; evidence for the efficacy of some algae extracts has been achieved also against *Colletotrichum* spp. For instance, BDDE [Bis(2,3-dibromo-4,5-dihydroxybenzyl) ether], derived from *Leathesia nana*, *Rhodomela confervoides*, and *Rhodomela confervoides*, was found effective in reducing the mycelial growth of *C. gloespoiriodes* species complex in vitro [114]; similarly, aqueous and ethanolic fractions extracted from the species *Sargassum myricocystum* and *Gracilaria edulis* exhibited a similar inhibitory activity against *C. falcatum* [115]. Conversely, the germination of *C. lindemunthianum* was reduced by a protein fraction extracted from *Hypnea musciformis* [116].

However, as for plant derivates and EOs, in vivo studies are necessary to move from laboratory tests to application in anthracnose management protocols. For example, “Ulvan”, a water-soluble polysaccharidic extract achieved from *Ulva* spp., was shown very effective as a resistance inducer in bean plants, resulting in a consistent (up to 60%) reduction of severity of *C. lindemuthianum* anthracnose [117,118]. An ethanolic fraction of the same species applied through foliar spray or infiltration was instead associated to an increased expression of defense gene markers in alfalfa specimens and to a subsequent enhanced resistance against *C. trifolii* infections [119]. Similarly, Kim et al. [120] pointed out that foliar treatments of cucumber plants with a *Chlorella fusca* suspension efficiently triggered the endogenous defense mechanisms, leading to a reduced severity of anthracnose caused by *C. orbicolare*. Finally, the progression of anthracnose caused by *C. acutatum* species complex on strawberry leaves was consistently limited through spray treatments with an *Ascophillum nodosum*-based biofungicide [121] (Table 3).

In general, the introduction of plant-based compounds into crop defense strategies, requires the fulfillment of a number of constraints besides the lack of adverse effects and the development of appropriate delivery techniques. Indeed, to obtain the authorization from the regulatory authorities, a complex dossier with data on their stability—presently not available for the majority of these compounds—must be developed [93,97,122]. In addition, the activity of these compounds seems to be not enough reproducible, mostly because of the wide variability of their chemical profile [97,98,123]. Therefore, despite the promising results achieved, the introduction of these substances into the market seems still far off.

### 3.2. Biotechnology-Based Strategies

In recent years, innovative biotechnology-based methodologies for pathogen and pest management have been considered as tools to increase plant resistance against pathogens or to develop new molecules alternative to pesticides.

One promising approach is genome editing that allows achieving new resistant varieties. This technique relies on the production of breaks in specific sites of the plant genomic DNA that are subsequently restored by different cell repairing mechanisms; through the process, single mutations (insertions, deletions or substitutions) can be introduced into the target loci [124]. Meganucleases (MNs), zinc finger nucleases (ZFNs), transcription activator-like effector nucleases (TALENs), and clustered regularly interspaced short palindrome repeats protein 9 (CRISPR-Cas9) are four types of nucleases that can be used in genome editing for DNA break production at specific locations. The great advantage of this technique, included in the so called “New Breeding Technologies” (NBT), is given by both precise and specific genomic modifications. These can target plant genetic traits associated with susceptibility to pathogens in order to strengthen the plant immune response. Compared to traditional breeding methods, these approaches greatly reduce the time required for developing new varieties with the traditional breeding technologies.

On the other hand, the application of this approach can be limited by the scarce knowledge on the genetic mechanisms regulating plant response to pathogens, which, in case of *Colletotrichum* are not totally uncovered. To date, the only scientific evidence on the efficacy of NBT application against this disease was reported by Mishra and colleagues [125] who developed CRISPR-Cas9 T-DNA-free homozygous pepper plants (*Capsicum annum*) harboring a desired mutation in the CaERF28 gene. The expression of this gene leads to the downregulation of the endogenous defense mechanisms [126,127] and transformed plants were significantly less susceptible to *C. truncatum*.

#### 3.2.1. Bidirectional Cross-Kingdom RNA Interference

Among the biotech-based technologies, RNA interference-based strategies have been more extensively explored than genome editing as putative next-generation tools for a sustainable crop protection.

RNA interference (RNAi) is a regulatory system of post-transcriptional gene silencing widely conserved among Eucaryotes and involved in several processes including host immunity, pathogen virulence, and host–pathogen communication [128,129,130,131,132,133,134]. Besides the pathogenesis processes, RNAi in fungi is engaged in multiple functions such as the control of transposable elements, regulation of endogenous gene transcription, heterochromatin formation, maintenance of genome stability, and adaption to stressful conditions [135]. 

The RNAi mechanism is mediated by small (21–26 nucleotides) RNA (sRNA) molecules with sequence complementarity to transcripts encoded by the target genes. sRNAs are obtained from the cleavage of longer double-stranded RNA (dsRNA) molecules, performed by Dicer or Dicer-like (DCL) proteins or sRNA-specific RNase III family enzymes. One strand of the cleaved dsRNA, so called the “guide” strand, is then incorporated into the RNA-induced silencing complex (RISC) by binding to the Argonaute (AGO) protein, while the second strand is degraded. The guide strand acts as a probe for the specific recognition of target transcripts enabling their cleavage by the RISC complex and preventing transcription [136,137,138].

In the plant kingdom, the RNAi mechanism plays an important role in defense against biotic stressors, downregulating endogenous genes involved in the host susceptibility [134]. In addition, it has been revealed that during the infection process, plant sRNAs can be transferred to pathogens in order to silence genes critical for their growth or virulence; similarly, pathogens can enhance their virulence by transferring to sRNA targeting host defense genes [133]. Altogether, this sRNA-based communication process, first identified in plant-fungi interactions [139], has been called “Bidirectional cross-kingdom RNAi” [131,132,139,140,141] and its discovery has represented a milestone for inspiring new strategies for crops protection [131,132,140,141]. Recent investigations on the mechanisms of sRNAs uptake by fungal pathogens from plant cells indicate that sRNA molecules are delivered from plants through exosomes or extracellular vesicles that are absorbed by fungal cells through endocytic fusion with the cell membrane [142,143,144,145]. 

RNAi thus opened up a new promising way to develop efficient, specific, and safe crop protection strategies against pests and pathogens. A fundamental prerequisite for its application is the presence of the RNAi machinery in the target organisms, not found in all the fungal pathogens species [135,145,146,147,148,149].

RNAi-based protection can be conferred in two ways: (i) host-induced gene silencing (HIGS) and (ii) spray-induced gene silencing (SIGS). Both ways can represent a valid alternative to other strategies traditionally adopted in the agri-food sector, however, some issues remain to be addressed for their field application (Figure 3).

#### 3.2.2. Host-Induced Gene Silencing Approaches

The HIGS strategy is based on the development of genetically modified crop plants expressing sRNAs (or longer dsRNAs, processed into sRNAs by the RNAi machinery) [150,151]. These sRNAs specifically target pathogen key genes arresting the infection processes and conferring a stable protection of the host [139,152,153,154]. Alternatively, sRNAs can be designed to silence plant genes encoding susceptibility factors, i.e., proteins preventing pathogen recognition and the host defense response [141,155].

This strategy has been demonstrated to be effective to protect crops from a large variety of pathogens and pests, including viruses, viroids, insect, and nematodes [156,157,158]. In 2017, the U.S. Environmental Protection Agency (EPA) approved transformed corn produced by Bayer by means of the SmartStax® PRO technology which make plants capable to express dsRNA molecules targeting Western corn rootworm transcripts [156,159].

HIGS technologies has been used in different crops to control a variety of fungal species as shown by several scientific reports [151,160,161,162,163,164,165]. Among these, Mahto et al. published in 2020 [166] the first evidence for the control of anthracnose in chilly and tomato. Here, the gene encoding the Conidial Morphology 1 protein (COM1), homolog to a *Magnaporthe oryzae* gene reported to have a crucial role in the conidium morphology and appressorium formation [167], was chosen as a target to control *C. gloeosporioides* species complex. *Agrobacterium*-mediated transformation of tomato and chilli plants with a *C. gloeosporioides*
*CgCOM1* sRNA expressing cassette resulted in a significant reduction of anthracnose symptoms in both leaves and fruit. These results were corroborated by microscopy analysis indicating the fungal growth inhibition and impairing the production of functional appressoria.

Despite the promising results achieved, the development of a HIGS strategy requires high investments in terms of time and cost necessary to generate stably transformed plants. Moreover, HIGS implies the generation of GMO plants, the use of which is limited by social concerns and national regulations.

#### 3.2.3. Spray-Induced Gene Silencing Approaches

SIGS technique is based on the exogenous application of dsRNAs targeting key transcripts of pathogens, and for this probably represents a more promising candidate to replace agrochemicals [141,153,156,168]. In recent years, several reports showed the efficacy of SIGS technologies for controlling fungal diseases, paving the way for the launch of new GMO-free RNAi-based crop protection strategies [129,163,167,168,169,170].

The efficiency of these techniques relies on dsRNA uptake by the pathogen, which can occur through direct absorption from the environment (the so-called environmental RNAi), or indirectly through the host (cross-kingdom RNAi) [131,132,171,172,173,174,175].

Despite many fungal species have been shown to efficiently uptake RNAs molecules from 21 up to 800 bp, SIGS through environmental RNAi do not work for all fungi, suggesting that some species are reluctant to absorb RNA, possibly due to biochemical feature of cell or membrane wall or the lack of essential RNAi machinery gene or protein component [131,134,169,171]. Thus, the application of this approach requires careful preliminary analysis of the target species.

The development of methodologies suited to increase the stability and persistence of dsRNA molecules in the field is another important requirement for SIGS application, since it is well-known that naked RNA undergoes rapid degradation in the environment. The use of carrier molecules, such as double-layered hydroxide or chitosan/carbon dot-based nanoparticles can be used to stabilize, increase durability, and help plant absorption of RNA [176,177,178,179,180,181,182,183,184]. 

Despite these aspects, the SIGS strategy remains a promising technology to substitute chemical pesticide application, given both the low toxicological profile of RNA and its low environmental persistency [185,186]. In addition, since RNAi silencing effect is based on the recognition of specific transcript sequences, the risk of the off-target effects is putatively very low. Contrary to the HIGS, the SIGS approach, besides being GMO-free, can be applied also on fruit at post-harvest stage [11,141].

Evidence of successful applications of SIGS against *Colletotrichum* spp. were first reported by Gu et al. in 2019 [187]. The silencing effect induced by dsRNAs recognizing a fragment of the *Fusarium asiaticum* gene *β2-tubulin* (Fa*β2tub-3*), highly conserved among fungi, was evaluated in different pathogenic species, including *C. truncatum*, the causal agent of anthracnose on soybean plants, where it efficiently inhibited the spore germination. Cross-kingdom RNAi was the mechanism suggested to be involved there, since the preliminary spray of dsRNA on susceptible host tissues, followed by later pathogen inoculation, led to conidial germination and mycelial growth inhibition. 

A functional RNAi machinery was recently reported for *C. abscissum* species [149], the causal agent of anthracnose on citrus fruits, whose growth was heavily impaired when a fungal strain was transformed so to express dsRNA targeting succinate dehydrogenase transcription. Conidial suspensions of mutant fungi were highly affected in germination and appressorium formation. Furthermore, the fungal inoculum did not determine relevant symptoms when infected into susceptible citrus tissues.

As stated above, SIGS strategies for crop protection can exploit but also the environmental RNAi process that implies the pathogen capability of external RNA uptake. On the other hand, when approached as environmental RNAi, SIGS was not an effective strategy to control *C. gloeosporioides* species complex, since fungal cells were not capable to uptake fluorescein-labelled dsRNA even after 20 h from the treatments [171]. Consistently, the topical application of dsRNA targeting the *DCL* gene of this species did not reduce the symptom occurrence in susceptible plant tissues, supporting the lack of dsRNA uptake ability in this species [153].

## 4. Concluding Remarks

The increased social concern about food safety and the environmental impact of the agri-food system, together with the forthcoming restrictions on the use of phytosanitary products, encouraged the research of innovative strategies alternative to toxic chemicals for crop protection. In this review the most promising strategies under investigation to control anthracnose in postharvest disease management are described. A number of issues must be addressed to allow their application in large-scale production systems. With respect to biocontrol agents, for instance, several investigations showed that a multitude of microorganisms are potentially suitable for anthracnose control; however, for the majority of cases, the antagonistic activity was explored only on few fruit systems and more studies are needed for developing commercial preparations [73]. In addition, their safety, absence of toxic effects and stability (i.e., capability of adhering and colonizing plant tissues, survival under adverse conditions, broad spectrum of action) should also be carefully addressed [70,73,188,189,190]. 

As about plant derivates, their antifungal activity was tested mostly on cultured fungi, whereas tests in relevant applicative environment are missing. Several EOs showed good efficacy to control anthracnose; however, further research is necessary to exclude phytotoxic effect on produces, to fulfil the requirements needed, for authorization and develop appropriate methodologies for their delivery. Nanotechnology is currently the more promising strategy to enhance their persistence and bioavailability in the environment taking into account also the economic and environmental sustainability of delivery formulations [97,181]. 

New strategies for *Colletotrichum* spp. management have been proposed also through biotechnology-based approaches, such as those exploiting the RNA interference mechanism (Table 4). 

Despite some promising results recently achieved [166] and the putative environmental safety of these methodologies, RNAi-based approaches still need to meet a number of technological, legal, and social issues. For instance, HIGS strategy is achieved through the development of GMO crops expressing interfering RNAs with all the regulatory and social concerns that this implies, although here plant transformation does not lead to synthesis of new proteins but to sRNA molecules acting in a sequence-specific manner [190,191,192,193,194,195]. Furthermore, for many susceptible crops fast and cost-effective transformation protocols are not available yet, making it difficult to reach large-scale production of sRNA expressing plants. On the other hand, once these limitations are overcome, this technique can provide stable protection allowing a substantial decrease in the use of toxic agrochemicals, a great advantage especially for fruit producers of developing countries, where productions are often seriously affected by anthracnose and postharvest technologies are not advanced. 

GMO-linked constraints do not affect the SIGS strategy, where interfering RNAs are exogenously applied. However, more studies are needed to consolidate knowledge on the RNAi functioning and siRNA uptake mechanisms by *Colletotrichum* spp., mostly with respect to the different behavior so far reported for different species [149,171,187]. These considerations become particularly important for the postharvest management of fruit crops, where restrictions in the use of agrochemicals and consumers’ attention on fruit safety raise particular concern. 

Owing to the sequence recognition mechanism, both RNAi-based approaches are characterized by high specificity toward target pathogens; however, the lack of a multi-spectrum efficacy can represent a limiting factor, especially for protection of crops susceptible to multiple *Colletotrichum* species. For this reason, the target gene/sequence selection process should be focused on the detection of regions efficient for the silencing of discrete related pathogen species. 

## Figures and Tables

**Figure 1 plants-11-01856-f001:**
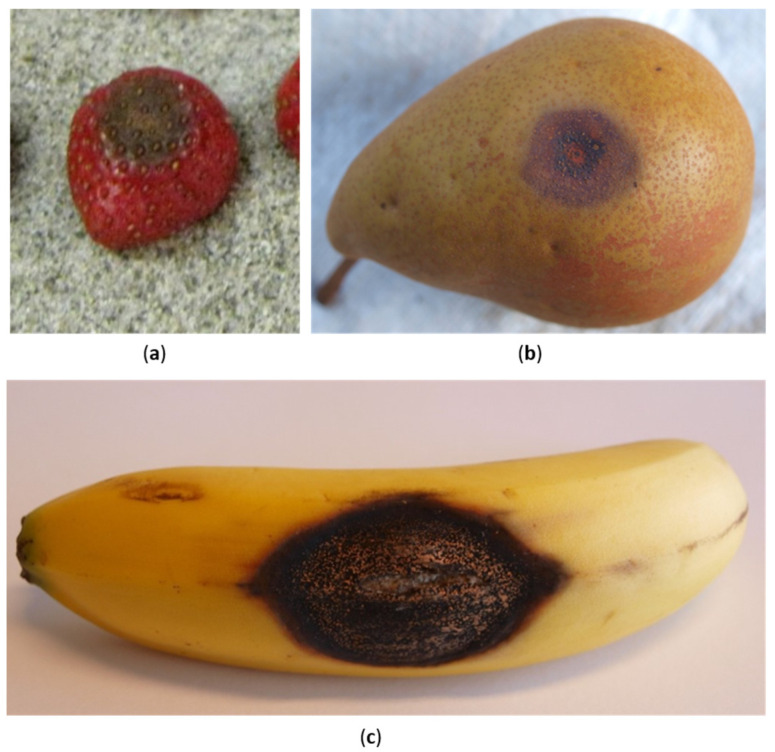
Anthracnose symptoms on harvested fruits. (**a**) Sunken brown lesions on strawberry fruits; (**b**) dark lesions surrounding orange conidia masses on pear fruits; (**c**) dark lesions on banana fruits.

**Figure 2 plants-11-01856-f002:**
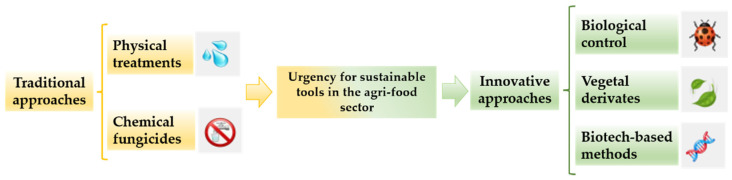
Approaches for post-harvest anthracnose management. Traditional strategies (yellow panels) and innovative methodologies explored for the improvement of the sustainability in the agri-food sector (green panels).

**Figure 3 plants-11-01856-f003:**
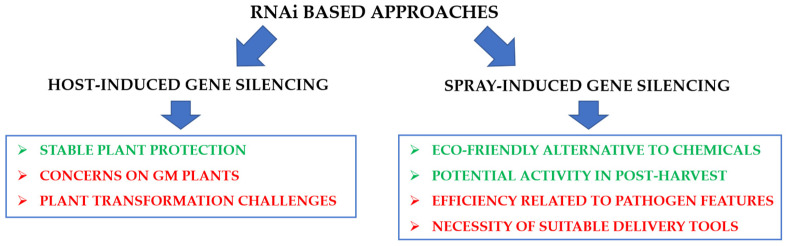
RNAi-based approaches for plant protection: host-induced gene silencing, HIGS (**left** panel) and spray-induced gene silencing, SIGS (**right** panel). The main advantages (in green) and disadvantages (in red) related to each strategy are reported. See text for further information.

**Table 1 plants-11-01856-t001:** Representative biological agents for *Colletotrichum* spp. management in post-harvest.

Biological Agent (Species and Strain/Isolate)	*Colletotrichum* Species	Host Species	Reference
*Bacillus subtilis* EA-CB0015	*C. acutatum* species complex	*Cyphomandra betacea*	[74]
*Bacillus subtilis* S16	*C. acutatum* species complex	*Malus pumila*	[75]
		*Prunus persica* cv. Chunjungdo	
		*P. persica* cv. Sunfre	
*Paenibacillus polymyxa* APEC128	*C. acutatum* species complex	*M. pumila*	[76]
	*C. gloeosporioides* species complex		
*Streptomyces philanthi* RM-1-138	*C. gloeosporioides* species complex	*Capsicum annuum*	[77]
*Burkholderia rinojensis* F2	*C. truncatum*	*C. annuum*	[78]
*B. rinojensis* F80			
*Burkholderia gladioli* F79			
*Burkholderia arboris* F35			
*Pseudomonas aeruginosa* F65			
*Trichoderma harzianum* T-39	*C. acutatum* sensu lato	*Fragaria* × *ananassa*	[79]
*Trichoderma hamatum* T-105			
*Trichoderma atroviride* T-161			
*Trichoderma longibrachiatum* T-166			
*T. harzanium* ^1^	*C. gloeosporioides* species complex	*Citrus limon* L.	[80]
*T. harzanium* TH-1	*C. musae*	*Musa acuminata*	[81]
*T. viridae* TV-3			
*T. viridae* TV-4			
*Trichoderma pseudokomngii* ^2^			
*Metchnikowia pulcherrima* FMB-24H-2	*C. acutatum* sensu lato	*Malus domestica*	[82]
*M. pulcherrima* T5-A2	*C. acutatum* sensu lato	*M. domestica*	[83]
*Pichia kluyveri* Y1125	*C. acutatum* species complex	*M. domestica*	[84]

^1^ Authors investigated pathogen inhibition in fruits from transgenic plants expressing a *T. harzanium* antifungal protein (see text for further information). ^2^ Strain/isolate not specified.

**Table 2 plants-11-01856-t002:** Representative essential oils agents for *Colletotrichum* spp. management in post-harvest.

Essential Oil	*Colletotrichum* Species	Host Species	Reference
Lemongrass oil	*C. gloeosporioides* species complex	*Carica papaya* L.	[99]
Ginger oil	*C. gloeosporioides* species complex	*C. papaya* L.	[100]
Savory oil	*C. gloeosporioides* species complex	*C. papaya* L.	[101]
	*C. gloeosporioides* species complex	*Persea americana*	[102]
Thyme oil	*C. gloeosporioides* species complex	*C. papaya*	[101]
	*C. gloeosporioides* species complex	*P. americana*	[101]
	*C. gloeosporioides* species complex	*Mangifera indica* L.	[103]
	*C. acutatum* species complex	*Fragaria* x *ananassa*	[104]
Cinnamon bark oil	*C. acutatum* species complex	*Fragaria* x *ananassa*	[104]
*Ocinum basilicum* oil	*C. musae*	*Musa* spp.	[105]
*Ocinum gratissimus* oil	*C. musae*	*Musa* spp.	[105]

**Table 3 plants-11-01856-t003:** Representative seaweed derivates for *Colletotrichum* spp. management.

Seaweed Derivate-Species	*Colletotrichum* Species	Host Species	Reference
“Ulvan”-*Ulva* spp.	*C. lindemuthianum*	*Phaseolus vulgaris* L.	[117]
“Ulvan”-*Ulva fasciata*	*C. lindemuthianum*	*P. vulgaris* L.	[118]
Ethanolic fraction-*Ulva* spp.	*C. trifolii*	*Medicago truncatula*	[119]
Algal suspension-*Chlorella fusca*	*C. orbiculare*	*Cucumis sativus*	[120]
Seaweed-based biofungicide-*Ascophyllum nodosum*	*C. acutatum* species complex	*Fragaria* × *ananassa*	[121]

**Table 4 plants-11-01856-t004:** Current investigations on the RNAi-based approaches for *Colletotrichum* management.

Approach	Outcome	Reference
HIGS	*C. gloeosporioides* species complex inhibition in GM plants carrying a CgCOM1 RNA expressing cassette	[166]
SIGS	*C. truncatum* silencing by means of interfering dsRNAs	[187]
	Existence of the RNAi machinery in *C. abscissum*	[149]
	Absence of dsRNA uptake efficiency in *C. gloeosporioides* species complex	[171]

## Data Availability

The data presented in this study are available in this article.

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
