# Peer review of "Management of Post-Harvest Anthracnose: Current Approaches and Future Perspectives"

_plants, 2022, doi:10.3390/plants11141856_

Round 1
Reviewer 1 Report
Dear authors,
the manuscript is very interesting. It is well organized and described and it represent an important contribution in the field.
Only a few suggestions are listed below:
In the section 1.2
Lines 61-65: It is very important underline the importance of disease in terms of economic losses and host-range. The authors could report other examples of Anthracnose diseases in post-harvest on other important crops, e.g. citrus, avocado, etc.
Line 91: Delete ‘See’ and use uppercase for ‘Figure 1’
Section 3.1.1
Lines 178-179: 'to reduce anthracnose incidence'. For which host-species?
Section 3.1.1
Lines 186-188: 'Despite this scientific evidence, to date, only two microorganism-based biofungicides against Colletotrichum spp. are commercially available'. Where? In the European Union?
Line 188: 'Serenade ASO (composed by B. subtilis 188 QST713'. The bacterial species is now classified as B. amyloliquefaciens QST713 (former B. subtilis)
Author Response
Dear referee,
we would like to thank you for your kind and careful revision to our manuscript entitled "Management of post-harvest anthracnose: current approaches and future perspectives". According to your comments and suggestions we modified our paper as follow:
Lines 61-65: It is very important underline the importance of disease in terms of economic losses and host-range. The authors could report other examples of Anthracnose diseases in post-harvest on other important crops, e.g. citrus, avocado, etc.: some examples were reported in lines 73-77 of the former version of the manuscript; in any case, we gave higher importance to this issues and we added other examples (see lines 79-86).
Line 91: Delete ‘See’ and use uppercase for ‘Figure 1’: we modified according your suggestion (see lines 104); the figure mentioned is the number 2 in the revised version of the manuscript.
Lines 178-179: 'to reduce anthracnose incidence'. For which host-species?: we modified according your suggestion (see lines 197-199).
Lines 186-188: 'Despite this scientific evidence, to date, only two microorganism-based biofungicides against Colletotrichum spp. are commercially available'. Where? In the European Union?: We indicated some countries where the microorganism-based biofungicides are commercially available (see lines 222-225).
Line 188: 'Serenade ASO (composed by B. subtilis 188 QST713'. The bacterial species is now classified as B. amyloliquefaciens QST713 (former B. subtilis): We modified the name of the species according to the updated classification (see lines 219-220).
Reviewer 2 Report
The paper titled “Management of post-harvest anthracnose: current approaches and future perspectives” is a great review about this topic. I think this manuscript contains a lot of useful and up-to-date information about this topic, which are synthesised by the authors. It is also useful to add a sustainable approach to solve the problem caused by anthracnose.
I have just some suggestions to improve the high quality of the paper. Is it possible to add some pictures about the symptoms; can be seen on the fruits during the post-harvest stage?
I think this manuscript will be read not just by scientist, but students, farmers etc. so it would be nice to add some basic sentences about the symptoms, which can be seen.
I think usage of the biological agents is the way of future. Do you have any data about the effectiveness of the mentioned biological agents? Are there any data about usage them in small outdoor orchards?
Author Response
Dear referee,
we would like to thank you for your kind and careful revision to our manuscript entitled "Management of post-harvest anthracnose: current approaches and future perspectives". According to your comments and suggestions we modified our paper as follow:
I have just some suggestions to improve the high quality of the paper. Is it possible to add some pictures about the symptoms; can be seen on the fruits during the post-harvest stage?: We added some images of fruits affected by anthracnose disease (see lines 73).
I think this manuscript will be read not just by scientist, but students, farmers etc. so it would be nice to add some basic sentences about the symptoms, which can be seen: We added to the images you suggested to insert to the manuscript a brief description of the symptoms (see lines 74-76).
I think usage of the biological agents is the way of future. Do you have any data about the effectiveness of the mentioned biological agents? Are there any data about usage them in small outdoor orchards? We added data on the efficacy of the biological agents (see lines 178-212), together with similar information about plants derivates (see lines 258-270). An interesting study reporting evidence for the effectiveness of a yeast species in mango orchard has been also indicated (see lines 212-214).